# Prophylactic Coloprotective Effect of *Urtica dioica* Leaves against Dextran Sulfate Sodium (DSS)-Induced Ulcerative Colitis in Rats

**DOI:** 10.3390/medicina59111990

**Published:** 2023-11-12

**Authors:** Nouha Dakhli, Kais Rtibi, Fatma Arrari, Ala Ayari, Hichem Sebai

**Affiliations:** Laboratory of Functional Physiology and Valorization of Bio-Ressources, Higher Institute of Biotechnology of Beja, University of Jendouba, Beja 382-9000, Tunisia; nouhadakhli88@gmail.com (N.D.); kais.rtibi@isbb.rnu.tn (K.R.);

**Keywords:** *Urtica dioica*, ulcerative colitis, oxidative stress, antioxidants, phenolic compounds

## Abstract

*Background and Objectives*: *Urtica dioica*, a source of bioactive functional compounds, provides nutritional and gastrointestinal therapeutic benefits. This study attempted to investigate the prophylactic coloprotective action of an aqueous extract of *Urtica dioica* (AEUD) on dextran sulfate sodium (DSS)-induced ulcerative colitis (UC). *Materials and Methods*: Phenolic compounds, total sugar, and mineral levels were determined in AEUD. Then, AEUD at different doses (50, 100, and 200 mg/kg, BW, p.o.) and mesalazine (MESA) as a standard treatment (100 mg/kg, BW, p.o.) were given orally for 21 days. Acute colitis was induced by administering drinking water with 5% (*w*/*v*) DSS for 7 days. Body weight variation, fecal occult blood, and stool consistency were determined daily. The severity of colitis was graded according to colon length, disease activity index (DAI), histological evaluations, and biochemical alterations. Rats orally administered DSS regularly developed clinical and macroscopic signs of colitis. *Results*: Due to its richness in phenolic and flavonoid compounds (247.65 ± 2.69 mg EAG/g MS and 34.08 ± 0.53 mg EQt/g MS, respectively), AEUD markedly ameliorated DAI, ulcer scores, colon length shortening, colonic histopathological changes, and hematological and biochemical modifications. Taken together, AEUD treatment notably (*p* < 0.01) suppressed DSS-induced UC by reducing oxidative stress via lowering MDA/H_2_O_2_ production and stimulating the effect of enzyme antioxidants as well as attenuating inflammation by decreasing CRP levels by 79.5% between the DSS and DSS + AEUD-50 groups compared to the MESA group (75.6%). *Conclusions*: AEUD was sufficient to exert a coloprotective effect that might be influenced by its bioactive compounds’ anti-inflammatory and antioxidant capabilities.

## 1. Introduction

Ulcerative colitis is one of the main forms of inflammatory bowel disease (IBD) affecting the rectum and colon that is characterized by an imbalance in proinflammatory and anti-inflammatory reactivity [1,2]. The remarkable rise in IBD worldwide, including in most developing nations, affects millions of individuals and is a major public health issue that may raise the risk of colon cancer [3,4]. Several factors contribute to the etiology of UC including microbial, environmental, and genetic factors [5].

Although the exact pathogenesis of UC is still unknown [6], it is characterized by relapsing and remitting mucosal inflammation [7,8]. In particular, macrophages play a significant role in inflammatory disorders by engendering the cytokines interleukin (IL)-1β and tumor necrosis factor-α (TNF-α) and other inflammatory mediators such as nitric oxide (NO) and prostaglandins [9]. Chronic inflammation has been linked to a lower risk of colitis-associated colorectal cancer by increasing the production of pro-inflammatory cytokines such as IL-6, IL-1ß, IL-17, and TNF-α. Therefore, targeting NF-kB inflammation pathways together with Wnt/ß-catenin signaling may act to control colorectal carcinogenesis [10]. In fact, it has been reported that the negative regulation of the Wnt signaling pathway by the degradation of β-catenin, a transcriptional coactivator of the Wnt signaling pathway, allows its subsequent translocation to the nucleus and activation of Wnt target genes by associating with LEF-1/TCF proteins [11].

The standard treatment methods for UC use medications targeting inflammation and the immune system including mesalamine, sulfasalazine, and corticosteroids that, taken alone or in association, contribute to treating this disease [12]. Unfortunately, these drugs are linked with side effects and patients eventually become refractory or intolerant over time [13]. Therefore, the research for alternative and/or supplementary treatments among herbal and traditional medicines has been highly motivated [14,15]. Recent studies have focused on natural products and supplements obtained from plants with anti-inflammatory effects, low toxicity, and minimal side effects [16].

*Urtica dioica*, often known as common nettle, is one of the most commonly used medicinal plants in the world due to its biologically active compounds [17]. The leaves of this plant have been reported to show hypotensive, anti-inflammatory, hypoglycemic, analgesic, anti-ulcer, antioxidant, antimicrobial, cytoprotective, and anticancer activities [18,19]. Some of the chemicals in this plant include lignan, secolignan, norlignan, alkaloid, sesquiterpenoid, flavonoid, triterpenoid, sphingolipid, and sterol [20,21]. The trichomes of the nettle contain formic acid, acetyl choline, serotonin, and histamine [22].

According to various studies, the stinging nettle plant contains biologically active chemicals such as phenols and flavonoids that can help reduce free radical generation by diverse pharmacological properties such as antioxidative and anti-inflammatory properties and may play a role in the prevention of intestinal inflammation. The use of water as a solvent showed the highest total phenolic content values as well as producing a significant effect on the antioxidant capacity of the extracts [23].

The aim of this study was to evaluate the prophylactic coloprotective action of AEUD on DSS-induced ulcerative colitis via the regulation of inflammatory reactions and antioxidant properties in a colitis rat model.

## 2. Materials and Methods

### 2.1. Animals

Healthy adult male Wistar rats (weighing between 180 and 200 g) were purchased from the Society of Pharmaceutical Industries of Tunisia (SIPHAT, Ben-Arous, Tunisia) and acclimatized for 1 week before performing any experiment. All animals were housed under safe laboratory conditions in a temperature- and humidity-controlled room (22–24 °C, 70%) and kept on a 12 h light/dark cycle using hygrometer, thermometer, and timer settings with food and tap water available ad libitum. All animal procedures were performed in accordance with the Guidelines for Care and Use of Animals Laboratory and approved by the Bio-Medical Ethics Committee (CEBM) for the Care and Use of Animals for scientific purpose (JORT454002 (6 May 2021)). Furthermore, all experiments were performed at the same time of day (8 h).

### 2.2. AEUD Preparation

Leaves of *Urtica dioica* were collected from Beja, Tunisia, in March 2021 and were identified by Dr. Chokri Hafsi, a Professor at the University of Jendouba. The Voucher specimens (No. SO.325) have been deposited with the herbarium of the Higher Institute of Biotechnology of Beja, Tunisia. After drying in an oven at 50 °C for 48 h, the leaves were ground into fine powder using a blender. An amount of 10 g of the powder mixture was dissolved in 100 mL of bi-distilled water and incubated in a shaker for 24 h. Then, the extract solution obtained was filtered, concentrated in a water bath under vacuo, frozen, and lyophilized. AEUD was used for the phytochemical and mineral determination and in vivo experiments.

### 2.3. AEUD Phytochemical and Mineral Analysis

A phytochemical characterization of AEUD was made by determining the total phenolic compounds according to the colorimetric method of Folin–Ciocalteu. Briefly, 500 µL of the extract was added to 10 mL of water and 0.5 mL of Folin–Ciocalteu reagent. After 5 min, 8 mL of 7.5% sodium carbonate solution was added. The reaction was kept in the dark for 2 h and was measured at 765 nm using a UV-visible detector spectrophotometer. Gallic acid was applied as a standard, and the results were expressed in milligram gallic acid equivalent per gram dry matter (mg GAE/g DM) [24].

The extract solution (0.5 mL) was mixed with 500 μL of 50% Folin–Ciocalteu reagent. The mixture was then allowed to stand for a 2–5 min period followed by the addition of 1.0 mL of 20% sodium carbonate. After 10 min incubation at room temperature, the mixture was centrifuged for 5 min (1000× *g*), and the absorbance of the supernatant was measured at 730 nm. The total tannin content was expressed as mg of tannic acid equivalents/g DM [25].

The total flavonoid content was detected using the AlCl_3_ colorimetric method. In fact, 1 mL of the sample was mixed with 1 mL of 2% AlCl_3_ solution. After 15 min incubation at room temperature, the optical density of their action mixture was evaluated at 430 nm. Quercetin was used as a reference standard and the total flavonoid content was expressed as milligram quercetin equivalent per gram dry matter (mg QE/g DM) [26]. The total sugar level was determined using a previous procedure [27].

Atomic spectroscopy was used to detect the contents of magnesium (Mg), zinc (Zn), iron (Fe), manganese (Mn), molybdenum (Mo), and copper (Cu) in AEUD.

### 2.4. Experimental Procedure

The study was continued for 21 days and a total of 36 rats were divided into six groups, each consisting of six animals, including: Group 1: normal control given only saline solution with oral intake of NaCl (0.9%, 5 mL/kg, b.w.); Group 2: the colitis group receiving DSS (5%) in the drinking water; Group 3: the reference group, MESA was administered to the rats at 100 mg/kg by gavage from day 0 to 21; Groups 4, 5, and 6: AEUD given at 50, 100, and 200 mg/kg once a day by gavage route for 21 days.

Ulcerative colitis was induced in rats by administering 5% DSS in the drinking water from day 15 to 21, except for the control group. During DSS treatment, stool consistency, the presence of blood in the feces, body weight, and food intake were examined and documented daily. After 21 days of experiment, animals were anesthetized to avoid any kind of stress which could distort the results and sacrificed by decapitation. The entire colon was measured; then, a portion of the colon tissue was stored in 10% buffered formalin for histopathological analysis and the remaining colon tissue was stored at −80 °C for further biochemical analysis.

### 2.5. Evaluation of Clinical Colitis and Colonic Weight and Length Measurement

The assessment of clinical colitis included daily monitoring of disease activity score determined on the basis of stool consistency, blood in the stool, and weight loss during exposure to DSS. The relevant specific criteria that were used to calculate the DAI are presented in Table 1 [28]. The samples of the large intestine were weighed and the colon lengths were measured.

### 2.6. Biochemical Assays

Blood samples were collected in lithium heparin tubes and then plasma was obtained by centrifugation (4000 t/min/4 °C for 15 min) and stored at −80 °C until analysis. Plasma levels of C-reactive protein (CRP), amylase, aspartate aminotransferase (AST), alanine aminotransferase (ALT), lactate dehydrogenase (LDH), alkaline phosphatase (ALP), cholesterol (TC), triglyceride (TG), high-density lipoprotein (HDL), low-density lipoprotein (LDL), blood sugar, urea, and creatinine were measured using automated enzymatic assays (ProXL). Potassium (K^+^) and sodium (Na^+^) were measured using a Cornley AFT-300 Electrolytes Analyzer (Precimed, China).

### 2.7. Determination of Hematological Parameters

Hematological parameters including hemoglobin (Hb), hematocrit (Hct), red blood cells (RBCs), white blood cells (WBCs) as well as hematological indices such as mean corpuscular volume (MCV), mean corpuscular hemoglobin (MCH), and mean cellular hemoglobin concentration (MCHC) were commonly analyzed using an electronic automate (HORIBA-ABX Pentra XL 80 (Bioplus, China)).

### 2.8. Assessment of DSS-Induced Oxidative Stress in the Colonic Tissues

Colonic lipid peroxidation was determined by measuring MDA using the double heating method. In brief, aliquots of colonic homogenates were mixed with a BHT–TCA solution that contained 1% BHT (*w*/*v*) dissolved in 20% TCA (*w*/*v*) and centrifuged at 1000× *g* for 5 min at 4 °C. The supernatant was mixed with 0.5 N HCl and 120 mM TBA in 26 mM Tris and heated at 80 °C for a duration of 10 min. The absorbance of the resulting chromophore was determined at 532 nm after cooling. The levels of MDA were determined using an extinction coefficient for the MDA–TBA complex of 1.56 × 105 M^−1^ cm^−1^ [29].

For the determination of glutathione peroxidase (GPx), 1 mL of reaction mixture containing 0.2 mL colonic homogenate supernatant, 0.2 mL (0.1 M) phosphate buffer pH 7.4, 0.2 mL GSH (4 mM), and 0.4 mL H_2_O_2_ (5 mM) was incubated at 37 °C for 1 min and the reaction was stopped by addition of 0.5 mL TCA (5%, *w*/*v*). After centrifugation at 1500× *g* for 5 min, an aliquot (0.2 mL) of the supernatant was combined with 0.5 mL of 0.1 M phosphate buffer pH 7.4 and 0.5 mL of DTNB (10 mM) and the absorbance was read at 412 nm. The GPx activity was expressed in nanomolar of GSH consumed per minute per milligram of protein.

The superoxide dismutase (SOD) activity level was obtained using modified epinephrine assays. At alkaline pH, the superoxide anion (O_2_^−^) causes the auto-oxidation of epinephrine to adrenochrome while competing with this reaction: SOD decreases the formation of adrenochrome. An SOD unit is the quantity of extract that inhibits the rate of adrenochrome formation by 50%. The enzyme extract was added to 2 mL reaction mixture containing 10 µL bovine catalase (0.4 U/µL), 20 µL epinephrine (5 mg/mL), and 62.5 mM sodium carbonate/ bicarbonate buffer pH 10.2. Absorbance changes were observed at 480 nm [30]. 

The catalase (CAT) activity was evaluated by measuring the initial rate of hydrogen peroxide (H_2_O_2_) disappearance at 240 nm. The reaction mixture contained 33 mM H_2_O_2_ in 50 mM phosphate buffer at pH 7.0 and the activity of CAT was calculated using the extinction coefficient of 40 mM^−1^ cm^−1^ for H_2_O_2_ [31,32].

The colonic mucosal H_2_O_2_ level was determined. Briefly, hydrogen peroxide reacts with p-hydroxybenzoic acid and 4-aminoantipyrine in the presence of peroxidase leading to the formation of quinoneimine that has a pink color detected at 505 nm [33].

### 2.9. Histopathological Analysis

Colonic tissue specimens from the distal portion were collected, washed with ice-cold saline solution, and fixed in phosphate-buffered formalin (10%). Then, specimens were embedded in blocks of paraffin, sliced into 3 to 5 μm sections, stained with hematoxylin and eosin (H&E), and assessed for mucosal damage, ulceration, erosions, hemorrhage, and necrosis by a pathologist in a blinded manner under light microscopy equipped with a color video camera for digital imaging [34].

### 2.10. Statistical Analysis

All values were evaluated as mean ± standard error of the mean. The statistical significance of differences between groups was measured using SPSS statistical program software version 20 using one-way analysis of variance with post hoc Tukey’s multiple comparison test. A value of *p* < 0.05 was considered significant.

## 3. Results

### 3.1. Bioactive Compound Profile and Mineral Composition

Table 2 illustrates our obtained results; AEUD is an excellent source of functional compounds such as total phenolics (247.65 ± 2.69 mg EAG/g MS), total tannins (188.29 ± 2.94 mg ECat/g MS), flavonoids (34.08 ± 0.53 mg EQt/g MS), and sugars (32.3 ± 2.4 mg Eq Glucose/g MS).

Our outcomes showed that AEUD contained a high concentration of magnesium, iron, and zinc (Table 3). 

### 3.2. Effects of AEUD or MESA on Colitis Clinical Symptoms and Colonic Weight/Length

Rats in the negative control group exhibited variations in body weight compared to their body weight at day 15, and food and water intakes. The rats showed a decrease in the food and water intakes in the model group versus those in the negative control group, which clarifies the body weight variation. In this context, the body weight of DSS-induced colitis rats varied significantly between 5.60 ± 0.80 g and 2.4 ± 0.21 g between day 15 and day 21 when comparing the negative control to the DSS group, respectively. However, it notably recovered in both the 50 and 100 mg/kg AEUD-treated groups (4.57 ± 0.38 g and 4.62 ± 0.40 g, respectively) and the MESA group (4.52 ± 0.50 g). Pretreatment with AEUD or MESA also improved the accompanying symptoms, which returned to approach ordinary qualities, in comparison to the DSS group (Table 4).

The DAI score is a common parameter used for evaluating the severity of colitis. As expected, in the rats receiving DSS for a week (from day 15 to day 21) and compared with the normal control group, there was a significant increase in the DAI score of the DSS group indicating that this group showed substantial variation in body weight as well as stool consistency and bloody stool.

On the other hand, the DAI scores were decreased in rats pretreated with AEUD with different extract doses and MESA (Figure 1).

Shortening of the colon length and decreased weight are both indirect indicators of DSS-induced colitis severity. We assessed both colon length and weight in rats given DSS and/or AEUD/MESA. We found that the colon length/weight was dramatically decreased in the DSS-treated group contrasted to the control group. Rats given DSS + AEUD-50, DSS + AEUD-100, DSS + AEUD-200 (13.15 ± 0.58 cm, 12.83 ± 0.68 cm, 12.00 ± 1.09 cm), or DSS + MESA (14.87 ± 0.75 cm) showed a statistically significant decrease in the shortening of their colons in contrast with DSS (11.58 ± 1.29 cm) rats. Colon weight was higher in all groups given both DSS and AEUD or MESA when compared to rats given DSS alone (Table 4).

### 3.3. Histological Observation and Evaluation

Micro-photographs of the colon sections are presented in Figure 2. The histopathological analysis of the colons of the control group and the pretreated-with-AEUD (50 mg/kg) group rats showed crypts with a normal histological structure containing normal mucosal epithelial cells along with submucosal glands with no observable ulceration or inflammation.

The colon tissues of the DSS (5%)-treated group were characterized by moderate to severe mucosal and submucosal inflammation along with infiltration of inflammatory cells, edema, and some ulcerations compared to normal tissue sections.

There was no significant difference between pretreated groups with MESA, AEUD (100 mg/kg), and AEUD (200 mg/kg); they showed mild erosion and mild focal mucosal and submucosal inflammatory cell infiltration, which were less severe findings compared to the colitis group and effectively reduced the signs of inflammation and retained the structural integrity with minimal pathological damage in colon tissue.

### 3.4. Biochemical Measures

As shown in Table 5, the plasma levels of inflammation marker (CRP), amylase, urea, creatinine, LDH, AST, ALT, and ALP were significantly higher in the DSS group than in the control group. In fact, compared with the control group, the CRP level in the DSS-treated group was significantly enhanced from 0.37 ± 0.08 mg/L to 2.05 ± 0.15 mg/L. With treatment with various doses of AEUD, the level of this inflammatory marker was reduced significantly to 0.42 ± 0.05 mg/L but not dose-dependently. The results further showed that the amylase activity in the DSS group was greatly increased (1047 ± 40.66 U/L) compared to the normal group (225 ± 14.86 U/L). Compared with that in the DSS group, the activity of amylase in the DSS + AEUD groups showed an attenuation of the increasing amylase activity. The same actions were noticed following the administration of MESA. Compared with those in the control group, AST, ALT, and ALP levels in the DSS-treated group were significantly elevated. In the AEUD treatment groups, these levels were lower than those in the DSS group and close to those in the control group. Added to that, we observed that treatment with AEUD greatly reduced the glycemic rise from 7.87 ± 0.42 mmol/L in the group of rats treated with DSS to 5.48 ± 0.29 mmol/L in the group treated with the higher dose of AEUD (200 mg/kg, BW, p.o.). Despite this, our outcomes indicated that the remaining parameters such as TC, TG, HDL, LDL, sodium, and potassium were not significantly changed in the DSS, MESA, and AEUD-treated groups versus the control group.

### 3.5. Hematological Parameters

The amount of platelets was lower in the DSS group of rats (557.33 ± 23.54 mEq/L) compared to the control group (903.5 ± 55.86 mEq/L). Similar to MESA, AEUD at various concentrations significantly increased the platelet amounts in the blood and were close to those in the control group. Furthermore, compared with the control treatment, DSS significantly upregulated the WBC concentration from 10.2 ± 0.28 to 14.96 ± 0.47 mg/L. While similar to MESA, AEUD at various concentrations significantly downregulated the WBC blood level (Table 6). However, there were no improvements in Hb, Hct, erythrocytes, MCV, MCH, and MCHC. 

### 3.6. Effects of DSS and AEUD on MDA and H_2_O_2_ Accumulation

DSS administration significantly (*p* < 0.05) increased the mean MDA level (54.06 ± 3.03 pmoles/mg protein) compared to that in normal rats (19.33 ± 2.90 pmoles/mg protein). Animals treated with MESA (100 mg/kg, BW, p.o.) and AEUD, especially at a dose of 100 mg/kg, exhibited a significant decrease in mean MDA level compared to colitis animals (26.04 ± 2.27 and 25.13 ± 2.96 pmoles/mg protein, respectively). Moreover, the hydrogen peroxide level was found to be raised significantly (*p* < 0.05) in colitis rats (1.02 ± 0.01 µmoles/mg protein) when compared with the negative control group (0.44 ± 0.04 µmoles/mg protein). Rats administered AEUD at the three doses of 50, 100, and 200 mg/kg and MESA exhibited significantly decreased levels of H_2_O_2_ compared to colitis rats (Figure 3A,B).

### 3.7. Effects of DSS and AEUD on Antioxidant Enzyme Activities

Figure 4 A–C represents the effects of AEUD at diverse doses (50, 100, and 200 mg/kg) and MESA on SOD, CAT, and GPx antioxidant activities in the colon tissues of all groups of animals. DSS-induced UC animals demonstrated significant depletions in antioxidative enzyme levels (0.53 ± 0.007 U/mg protein, 2.55 ± 0.19 µmoles H_2_O_2_/mg protein, and 0.28 ± 0.05 µmoles GSH/min/mg protein, respectively), revealing a depletion in free radical scavenging activity compared to the normal control group (1.11 ± 0.03 U/mg protein, 6.54 ± 0.13 µmoles H_2_O_2_/mg protein, and 0.48 ± 0.03 µmoles GSH/min/mg protein, respectively). Animals treated with AEUD at various doses and MESA at a single dose of 100 mg/kg demonstrated significant replenishment in mean SOD, CAT, and GPx levels compared to UC animals, indicating restoration of free radical scavenging activity. Dose-dependence was noticed only when observing the SOD and GPx activities. 

## 4. Discussion

UC is an inflammatory disorder of the colon with a complicated etiology. 5-ASA, often known as mesalazine, is one of the most commonly used medicines in the treatment of UC. However, these treatments have the potential to have negative side effects. Consequently, there is growing interest in using the numerous natural components found in traditional herbs. In this respect, the prophylactic coloprotective action of the total extract of *Urtica dioica* leaves was investigated in DSS-induced UC in rats.

Firstly, we evaluated the phytochemical and mineral composition of AEUD which has been used as a natural remedy for ages [35]. The current results exhibit that AEUD is rich in bioactive compounds including phenolic compounds, tannins, flavonoids, and sugar. Furthermore, AEUD contains high mineral levels such as those of magnesium, iron, and zinc. These data are similar to those of previous studies which confirm that stinging nettle’s leaves are becoming more well-known because they contain a wide range of chemical components such as flavonoids, phenolic compounds, organic acids, minerals, and vitamins [36], as well as tannins, fatty acids, volatile compounds, polysaccharides, isolectins, sterols, terpenes, and proteins [37,38].

Secondly, the current study was designed to investigate the protective effects of AEUD in a DSS-induced ulcerative colitis model. The DSS–induced colitis model is one of the most widely used models. It can potentially cause damage to intestinal epithelial cells (IECs) [39]. The resulting injury has symptoms and characteristics similar to UC in humans [40]. In this regard, rats treated with DSS in their drinking water demonstrated a significant decrease in clinical parameters such as food intake, water consumption, body weight gain, and colonic length/weight due to severe tissue edema, necrosis, goblet cell hyperplasia, and inflammatory cell infiltration [41], but an increase in the DAI. MESA and AEUD treatment significantly improved these symptoms. Previously, many studies have shown the benefits of pure polyphenols or polyphenol-rich extracts in preventing colonic length/weight decreases in colitis rats, which is indicative of the therapeutic efficacy of prospective anti-ulcerative medicines [42,43].

DSS control rats had mild to severe mucosal and submucosal inflammation, inflammatory cell infiltration, edema, and some ulcerations as compared to normal tissue sections. On the other hand, the colonic tissues of the rats treated with AEUD (50 mg/kg) were determined to have normal histological structure.

When compared to the colitis group, the MESA or AEUD (100 mg/kg and 200 mg/kg) pretreated groups demonstrated mild erosion, mild focal mucosal inflammation, and mild submucosal inflammation. A previous investigation confirms that the administration of dextran alone does not result in any symptomatology in mice; it is actually the sulfate groups that are responsible for DSS toxicity [44]. In fact, DSS is responsible for altering the intestinal barrier integrity which disturbs the intestinal microbiome and homeostasis of intestinal immunity. The activation of the intestinal immune system and the migration of inflammatory cells into the intestine contribute to the maintenance of inflammation and intestinal lesions [45,46]. Many studies have demonstrated that the administration of several antioxidant agents reduces the severity of DSS-induced epithelial damage [47], mucosal inflammation, and erosion of surface epithelial cells [48,49].

AST and ALT levels were increased by DSS administration; however, AEUD reverted this rise in the DSS-treated rats. Cellular enzyme leakage into plasma is a well-known indicator of hepatic injury in conjunction with liver damage. Increased levels of these enzymes are reliable indicators of liver function because they show increased permeability, injury, and/or necrosis of hepatocytes [50].

CRP and WBCs are commonly used as markers of inflammation. We found an increase in these parameters in DSS-treated mice. In contrast, AEUD demonstrated great efficacy in preventing DSS-enhanced inflammatory mediators. In fact, CRP acts as an opsonin and activates complement, which causes the phagocytosis of bacterial and nuclear material. Therefore, CRP plays a crucial role in the innate immune system of the host and in the defense against autoimmunity [51]. High serum CRP levels in UC correlate well with disease activity and other inflammatory indicators such as WBCs. In a retrospective single-center cohort study, a higher WBC count at diagnosis was found to be associated with colectomy, underscoring the importance of WBCs in UC [52]. Many studies have found clinical cases of acute idiopathic pancreatitis and chronic pancreatitis related to IBD [53]. These findings demonstrated that pretreatment with AEUD or MESA reduced hyperamylasemia in DSS-treated rats. Increased permeability of the inflamed mucosa may be the cause of the pancreatic enzyme rise found in more severe or active disease; this is a mechanism previously suggested in persons with intestinal infarction who also have elevated serum amylase levels [54].

We found that AEUD significantly reduced the elevated glycemic level due to its ability to control blood sugar [55]. According to several studies, nettle enhances the release of insulin, which decreases blood sugar levels. This was demonstrated by examining diseased and healthy rats following intraperitoneal treatment with *Urtica dioica* aqueous extract [56].

The leaves of *Urtica dioica* have been reported to show anti-inflammatory properties and can be used to treat persistent inflammatory conditions. In this context, previous studies showed that biosynthesis of the arachidonic acid cascade enzymes, particularly the cyclooxygenases COX-1 and COX-2, was inhibited by leaf extracts, which reduced the formation of prostaglandins and thromboxane [57]. Furthermore, the PAF (platelet activating factor) system, inflammatory response, and antioxidant reaction are inhibited. The NF-κB system, which is implicated in immunity, is also affected [19,58]. In addition, a number of studies have demonstrated that leaf extracts block the release of interleukins IL-2 and IL-1, interferon (IFN), and the tumor necrosis factors (TNF) [59,60].

The ability of AEUD to combat oxidative damage and inflammation may be responsible for its protective action against liver injury related to oxidative damage [18]. Despite this, ROS are known to be beneficial species; however, excessive generation alters the redox balance and results in an oxidative stress state. In this context, it has been noted that an excessive amount of ROS is produced in subjects with UC. Hence, chronic oxidative stress has been demonstrated to have a significant impact on the persistence and etiology of ulcerative colitis [61]. In our investigation, exposure to DSS was accompanied by colonic oxidative damage that showed up as an increase in the H_2_O_2_ levels compared to the control group. Treatment with MESA and AEUD has proven to be useful in preventing colonic ROS excess caused by DSS intoxication. In fact, free radicals are molecules created by cellular metabolism, which can be destructive to biological tissues and cause injury to DNA, lipids, cell membranes, and proteins. It is commonly recognized that reactive oxygen species, especially the hydroxyl radical, contribute significantly to inflammation by causing membrane lipid peroxidation, which causes severe cellular damage [62]. Several studies have demonstrated that leaf extracts exhibit an antioxidant action by scavenging the DPPH radical (1,1-diphenyl-2-picrylhydrazyl). The majority of this antioxidant activity is caused by the presence of phenolic compounds [63,64].

One of the main causes of tissue lipid peroxidation is the excessive generation of ROS [65]. Our findings demonstrate that the colons of DSS control animals had higher MDA levels, whereas the administration of AEUD or MESA considerably decreased them. This shows that AEUD’s active ingredient has a protective impact that is remarkably similar to MESA’s protective effect in reducing oxidative damage. Our data are consistent with a number of previous reports that show that the antioxidant effect of AEUD is mostly due to the presence of phenolic compounds [63,64].

The activities of the main antioxidant enzymes SOD, CAT, and GPx were evaluated, since the occurrence of UC is significantly influenced by oxidative stress and inflammatory reactions [66]. SOD levels, GPx levels, and CAT activities dropped in the group treated with DSS, which could access mucosal cells via pinocytosis, causing cellular oxidation and disruption of the enzymatic antioxidant defense mechanism [67]. These levels increased and were comparable to those of the control group in the groups treated with AEUD and MESA. These outcomes are in accordance with the findings confirming that GPx, SOD, and CAT activities decreased in the colitis group under the action of the *Urtica dioica* antioxidant compounds [68,69,70,71].

## 5. Conclusions

The findings of this study demonstrate that AEUD improves colonic inflammation and oxidative stress in DSS-induced colitis by decreasing CRP levels, promoting the main antioxidant enzyme activities, and reducing colonic lipid peroxidation and H_2_O_2_ levels. Ultimately, AEUD alleviates DSS-induced colitis in rats and shows significant potential for bioactive compounds as a source of appropriate agents for the treatment of UC. However, further studies would help establish the local anti-inflammatory potential of the extract by evaluating the effect of AEUD on mediators of inflammation expressed in colonic tissues in colitis.

## Figures and Tables

**Figure 1 medicina-59-01990-f001:**
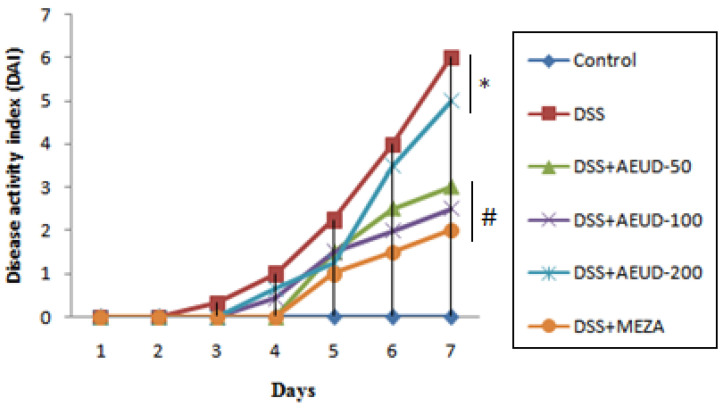
Effects of AEUD and MESA on disease activity index under colitis induced by DSS (from day 15 to day 21). Animals were pre-treated with three doses of AEUD (50, 100, and 200 mg/kg, b.w, p.o.), MESA (100 mg/kg, b.w, p.o.), or distilled water, and challenged with DSS (5%, *w*/*v*) in the drinking water or oral intake of NaCl (0.9%, 5 mL/kg, b.w.) during the last 7 days of treatment. Data are expressed as mean ± SEM (n = 6). * *p* < 0.05 compared to control group, # *p* < 0.05 compared to DSS group.

**Figure 2 medicina-59-01990-f002:**
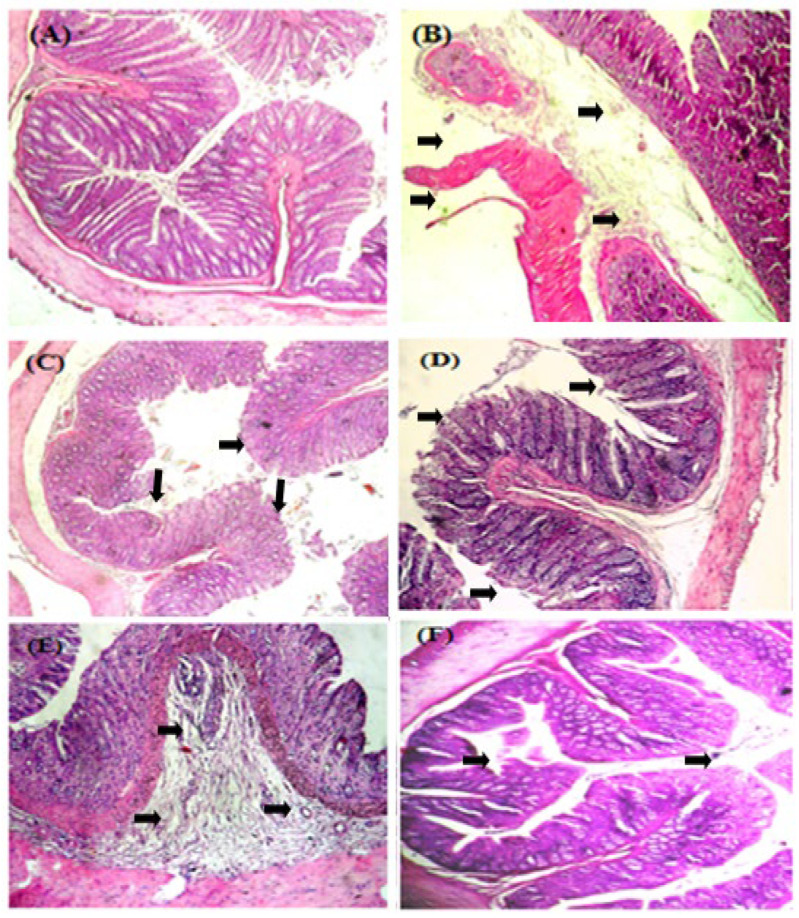
Effects of AEUD and MESA on the microscopic appearance of rat colons after DSS-induced colitis. Animals were pre-treated with various doses of AEUD (50, 100, and 200 mg/kg, b.w, p.o.), MESA (100 mg/kg, b.w, p.o.), or distilled water, and challenged with DSS (5%, *w*/*v*) or NaCl (0.9%, 5 mL/kg, b.w.) during the last 7 days of treatment. (**A**) H_2_O + NaCl, (**B**) DSS (5%, *w*/*v*) + H_2_O, (**C**) DSS + AEUD-50 mg/kg, (**D**) DSS + AEUD-100 mg/kg, (**E**) DSS + AEUD-200 mg/kg, (**F**) DSS + MESA (100 mg/kg, b.w., p.o.). The negative control group maintained normal colon morphology, whereas the DSS group showed multifocal areas of mucosal erosions with the loss of epithelial cells and inflammatory cell infiltration, edema, and ulceration (black arrows), (magnification ×40).

**Figure 3 medicina-59-01990-f003:**
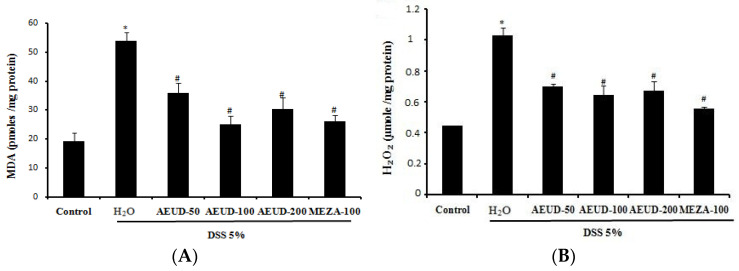
Effects of AEUD and MESA on dextran sulfate sodium (DSS)-induced changes in colonic mucosal MDA (**A**) and H_2_O_2_ (**B**) levels in rats. Animals were pre-treated with three doses of AEUD (50, 100, and 200 mg/kg, b.w, p.o.), MESA (100 mg/kg, b.w, p.o.), or distilled water, and challenged with DSS (5%, *w*/*v*) or NaCl (0.9%, 5 mL/kg, b.w.) during 7 days. The data are expressed as means ± S.E.M. (n = 6). *: *p <* 0.05 compared to control group. #: *p <* 0.05 compared to DSS group.

**Figure 4 medicina-59-01990-f004:**
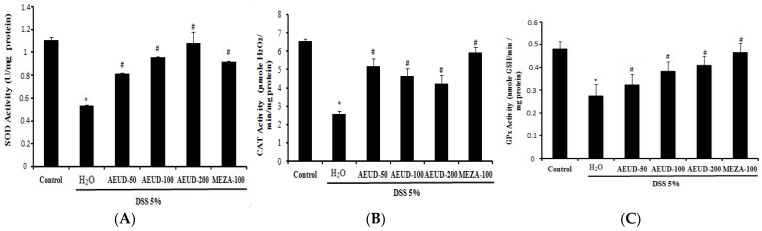
Effects of AEUD and MESA on colonic mucosal antioxidant enzyme activities: SOD (**A**), CAT (**B**), and GPx (**C**) during DSS–induced colitis in rats. Animals were pre-treated with three doses of AEUD (50, 100, and 200 mg/kg, b.w, p.o.), MESA (100 mg/kg, b.w, p.o.), or distilled water, and challenged with DSS (5%, *w*/*v*) or NaCl (0.9%, 5 mL/kg, b.w.) during 7 days. The data are expressed as means ± S.E.M. (n = 6). *: *p <* 0.05 compared to control group. #: *p <* 0.05 compared to DSS group.

**Table 1 medicina-59-01990-t001:** Disease Activity Index (DAI) score.

Score	Weight Loss	Stool Consistency	Bleeding
0	No weight loss	Formed	No bleeding
1	5–10%	Mild soft	Few blood-tinged stools
2	11–15%	Very soft	Slight bleeding
3	16–20%	Watery stool	Gross bleeding
4	>20%	-	-

**Table 2 medicina-59-01990-t002:** Phytochemical composition of AEUD.

Dietary Component	AEUD
Total phenolics ^a^	247.65 ± 2.69
Total tannins ^b^	188.29 ± 2.94
Flavonoids ^c^	34.08 ± 0.53
Total sugars ^d^	32.30 ± 2.40

Data are expressed as means ± standard error of the mean SEM (n = 5). ^a^ mg GAE/g DW; ^b^ mg QE/g DW; ^c^ mg QTE/g DW; ^d^ mg GE/g DW.

**Table 3 medicina-59-01990-t003:** Mineral composition of leaves of *Urtica dioica* (ppm).

Component	Leaves of *Urtica dioica*
Zinc	137.58
Copper	10.54
Manganese	42.92
Molybdenum	3.16
Iron	447.41
Magnesium	6845.97

**Table 4 medicina-59-01990-t004:** Effects of AEUD and MESA on DSS-induced changes in food intake, water consumption, body weight variation, and colon length/weight.

Parameters	Food Intake (g)	Water Consumption (mL)	Body Weight Variation (g)	Colon Length (cm)	Colon Weight (g)
Control	217.00 ± 4.08	89.57 ± 4.23	5.60 ± 0.80	16.07 ± 0.29	1.88 ± 0.15
DSS	172.42 ± 3.82 *	63.28 ± 2.56 *	2.4 ± 0.21 *	11.58 ± 1.29 *	1.34 ± 0.06 *
DSS + AEUD-50	190.00 ± 2.38 ^#^	79.00 ± 2.23 ^#^	4.57 ± 0.38 ^#^	13.15 ± 0.58 ^#^	1.64 ± 0.14 ^#^
DSS + AEUD-100	200.71 ± 2.13 ^#^	76.28 ± 3.86 ^#^	4.62 ± 0.40 ^#^	12.83 ± 0.68 ^#^	1.54 ± 0.05 ^#^
DSS + AEUD-200	200.42 ± 2.63 ^#^	67.85 ± 4.87 ^#^	3.57 ± 0.28 ^#^	12.00 ± 1.09 ^#^	1.50 ± 0.12 ^#^
DSS + MESA	203.71 ± 4.42 ^#^	85.00 ± 5 ^#^	4.52 ± 0.50 ^#^	14.87 ± 0.75 ^#^	1.81 ± 0.02 ^#^

Data are expressed as mean ± SEM (n = 6). * *p* < 0.05 compared to control group, ^#^
*p* < 0.05 compared to DSS group.

**Table 5 medicina-59-01990-t005:** Effects of AEUD and MESA on serum metabolic and biochemical parameters in diverse groups.

Groups	Control	DSS	DSS + AEUD-50	DSS + AEUD-100	DSS + AEUD-200	DSS + MESA
CRP (mg/L)	0.37 ± 0.08	2.05 ± 0.15 *	0.42 ± 0.05 ^#^	0.83 ± 0.16 ^#^	1.24 ± 0.16 ^#^	0.5 ± 0.08 ^#^
Amylase (U/L)	225 ± 14.86	1047 ± 40.66 *	831.66 ± 31.25 ^#^	854 ± 38.04 ^#^	825 ± 22.64 ^#^	785.5 ± 37.34 ^#^
Urea (mmol/L)	4.47 ± 0.33	6.46 ± 0.2 *	6.06 ± 0.36	6.17 ± 0.57	6.01 ± 0.29	6.02 ± 0.67
Creatinine (µmol/L)	37.5 ± 1.29	45.83 ± 1.6 *	45.16 ± 2.13	45.83 ± 3.06	46.4 ± 3.91	40.5 ± 2.42
AST (U/L)	85.75 ± 2.98	189.16 ± 15.94 *	89.2 ± 2.58 ^#^	91.5 ± 2.07 ^#^	94.75 ± 3.3 ^#^	105.33 ± 8.35 ^#^
ALT (U/L)	37.5 ± 2.64	59.66 ± 6.97 *	49.16 ± 1.16 ^#^	42.83 ± 3.18 ^#^	40.25 ± 2.87 ^#^	56 ± 3.89 ^#^
ALP (U/L)	59.75 ± 3.3	141.2 ± 4.08 *	91.6 ± 1.51 ^#^	86.4 ± 3.84 ^#^	88.6 ± 2.3 ^#^	84 ± 3.31 ^#^
LDH (U/L)	414.4 ± 22.46	971.83 ± 63.12 *	901.75 ± 4.34	949.4 ± 62.3	870.25 ± 31.48	666.33 ± 36.43
TC (mmol/L)	1.23 ± 0.33	1.24 ± 0.19	1.14 ± 0.34	1.14 ± 0.21	1.06 ± 0.08	1.17 ± 0.17
TG (mmol/L)	0.95 ± 0.07	1.005 ± 0.14	0.64 ± 0.08	0.62 ± 0.05	0.94 ± 0.06	0.86 ± 0.08
HDL (mmol/L)	0.59 ± 0.06	0.49 ± 0.04	0.5 ± 0.05	0.52 ± 0.04	0.5 ± 0.06	0.56 ± 0.06
LDL (mmol/L)	0.23 ± 0.04	0.32 ± 0.03	0.26 ± 0.04	0.3 ± 0.03	0.32 ± 0.04	0.22 ± 0.06
Glu (mmol/L)	7.16 ± 0.5	7.87 ± 0.42	5.35 ± 0.48 ^#^	5.69 ± 0.22 ^#^	5.48 ± 0.29 ^#^	7.09 ± 0.5
Sodium (mEq/L)	131.2 ± 0.83	137 ± 1.41	136.66 ± 2.94	136.16 ± 2.63	137.2 ± 3.34	135.66 ± 3.44
Potassium (mEq/L)	5.79 ± 0.48	6.1 ± 0.71	6.96 ± 0.86	6.96 ± 0.41	6.26 ± 0.53	5.75 ± 0.62

Data are expressed as mean ± SEM (n = 6). * *p* < 0.05 compared to control group, ^#^
*p* < 0.05 compared to DSS group.

**Table 6 medicina-59-01990-t006:** Effects of AEUD and MESA on hematological parameters in diverse groups.

Groups	Control	DSS	DSS + AEUD-50	DSS + AEUD-100	DSS + AEUD-200	DSS + MESA
WBC (mg/L)	10.2 ± 0.28	14.96 ± 0.47 *	10.75 ± 0.35 ^#^	9.6 ± 0.14 ^#^	10.35 ± 0.21 ^#^	10.23 ± 0.2 ^#^
Erythrocytes (U/L)	9.78 ± 0.24	9.57 ± 0.53	7.25 ± 0.8	7.4 ± 0.16	9.95 ± 0.07	11 ± 1
Hb (mmol/L)	15.4 ± 0.28	15.7 ± 0.7	15.05 ± 1.06	15.15 ± 0.07	15.9 ± 0.7	16.5 ± 1.05
Hct (µmol/L)	47.65 ± 1.34	47.3 ± 0.8	46.05 ± 4.73	45.8 ± 4.38	48.8 ± 0.28	50.63 ± 7.44
MCV (U/L)	49 ± 0.1	50 ± 1	51 ± 1.41	53 ± 0.6	50.5 ± 0.7	46 ± 2.64
MCH (U/L)	15.2 ± 0.1	16.4 ± 1.34	17.55 ± 0.49	18.65 ± 0.35	16.75 ± 0.21	14.93 ± 1.45
MCHC (U/L)	31.25 ± 0.07	32 ± 0.69	36.25 ± 0.91	36.05 ± 0.63	35.7 ± 0.28	32.5 ± 1.47
Platelets (mEq/L)	903.5 ± 55.86	557.33 ± 23.54 *	864 ± 32.52 ^#^	835 ± 77.78 ^#^	862 ± 56.56 ^#^	936 ± 49.92 ^#^

Data are expressed as mean ± SEM (n = 6). * *p* < 0.05 compared to control group, ^#^
*p* < 0.05 compared to DSS group.

## Data Availability

The data presented in this study are available on request from the corresponding author.

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
