# Peer review of "Prophylactic Coloprotective Effect of Urtica dioica Leaves against Dextran Sulfate Sodium (DSS)-Induced Ulcerative Colitis in Rats"

_medicina, 2023, doi:10.3390/medicina59111990_

Round 1

Reviewer 1 Report

Comments and Suggestions for Authors

The study by Nouha Dakhli et al,. Prophylactic coloprotective effect of Urtica dioica leaves against 2

dextran sulfate sodium-(DSS) induced ulcerative colitis in rat is creating an interests however I have some major revision is as follows-

1.     Authors have mentioned The severity of colitis was graded by colon length, need to provide a photographs of colons from different group to create a better clarity on severity scores.

2.     Why only adult male Wistar rats were chosen for evaluation? 

3.     Why authors have estimated the aspartate aminotransferase (ASAT), alanine aminotransferases (ALAT)?

4.     Mention the correct statistical anlysis and software details in 2.10. Statistical analysis section. Why tukey test was used?

5.     Rather mentioning Body weight variation (%), its better to have weight in gms for better readability.

6.     P value is not mentioned on DAI score

7.     Histopathology image is not clear to see nfiltration, edema, and ulceration, provide better images, I can’t see ulceration.

8.     Why the level of LDH increased in DSS?

9.     H2O2  level decreased in treatment groups? Is AEUD not involved to reduce oxidative stress?

10.   H2O2 and Catalase both were evaluated? Why? What is difference?

11.   Discussion needs to be written, many syntax were seen.

Comments on the Quality of English Language

Needs extensive language improvement

Author Response

Reviewer 1 :

  1. Authors have mentioned The severity of colitis was graded by colon length, need to provide a photographs of colons from different group to create a better clarity on severity scores.

We deeply appreciate the reviewer’s comment, but, the photographs of colons cannot be added to the manuscript since they are not preserved. 

  1. Why only adult male Wistar rats were chosen for evaluation?

We have chosen adult male Wistar because they have more stable hormonal status. During the reproductive cycle, the hormones fluctuations may influence the results with female rats, and we don’t have this problem with male rats.

  1. Why authors have estimated the aspartate aminotransferase (ASAT), alanine aminotransferases (ALAT)?

We estimated in our study the aspartate aminotransferase and alanine aminotransferase because we are interested to evaluate the effect of inflammation induced by DSS on liver status. These two enzymes are maken by liver, and in case of damage we can observe a rise in AST and ALT levels in blood.

  1. Mention the correct statistical anlysis and software details in 2.10. Statistical analysis section. Why tukey test was used?

This point has been clarified in 2.10. Statistical analysis section. 

  1. Rather mentioning Body weight variation (%), its better to have weight in gms for better readability.

This point has been corrected in the result’s part.

  1. P value is not mentioned on DAI score

Thank you for your comment, P value has been mentioned on DAI score

  1. Histopathology image is not clear to see nfiltration, edema, and ulceration, provide better images, I can’t see ulceration.

 The histopathology image has been changed

  1. 8. Why the level of LDH increased in DSS?

Lactate dehydrogenase (LDH) is an enzyme found in almost all the body’s tissues. So, when these tissues are damaged, they release LDH into the blood, and we obtain a rise levels which is the case with DSS which causes damaged in tissues.

  1. H2O2 level decreased in treatment groups? Is AEUD not involved to reduce oxidative stress?

Absolutely, H2O2 (representative of endogenous ROS, can be produced by almost all oxidative stress stimuli and oxidize various types of macromolecules: carbohydrates, nucleic acids, lipids, and proteins, thus, affecting the redox balance) level decreased in AEUD treatment groups which confirms its involvement to reduce the stress oxidative.

  1. H2O2 and Catalase both were evaluated? Why? What is difference?

The Fenton reaction, which occurs between H2O2 and Fe2+ ions to produce the extremely reactive OH radical, is regarded to be the primary mechanism of oxidative damage. However, catalase is a critical antioxidant enzyme that significantly reduces oxidative stress by eliminating cellular hydrogen peroxide to generate water. As a result, measuring these indicators is appropriate for assessing oxidative stress.

  1. Discussion needs to be written, many syntax were seen.

This is treated in the manuscript. Please see the discussion part.

Reviewer 2 Report

Comments and Suggestions for Authors

3.4.

As shown in Table 5, plasma levels of inflammation marker (CRP) -> C-reactive protein or inflammatory marker

ASAT, ALAT ???? - you mean AST, ALT 

compared to the normal group (225±14.86 U/L). Compared with that ...

- too many repeated words 

In many paragraphs word "compared" is repeated. Try to find a different word.

1. Introduction

"The exact pathogenesis of UC is still unknown[6], but, it is characterized by relapsing 36 and remitting mucosal inflammation" should be:

Although, the exact pathogenesis of UC is still unknown, it is characterized by relapsing 36 and remitting mucosal inflammation [6].

 "and other inflammatory mediators like nitric oxide 39 (NO) and prostaglandins" -> such as nitric oxide

After interpunction mark in many places it should be a blank space:

for example "Urtica dioica,often"

should be "Urtica dioica, often"

"become refractory or intolerant over time [13],and"

should be become refractory or intolerant over time, and  .... [13]

Too many old references.

Comments on the Quality of English Language

I think it requires some stylic, interpunction, and language editing.

Author Response

Reviewer 2:

As shown in Table 5, plasma levels of inflammation marker (CRP) -> C-reactive protein or inflammatory marker

C-reactive protein (CRP) is a protein maken by the liver and high levels of CRP may mean you have a serious health condition that causes inflammation. CRP test may be used to help inflammation in acute or chronic conditions, including ulcerative colitis, so it’s considered as a marker of inflammation.

ASAT, ALAT ???? - you mean AST, ALT

Aspartate aminotransferase (ASAT, AST or SGOT) and alanine aminotransferase (ALAT, ALT or SGPT), but it’s corrected in the manuscript by AST and ALT.

compared to the normal group (225±14.86 U/L). Compared with that ...

- too many repeated words

In many paragraphs word "compared" is repeated. Try to find a different word.

According to the reviewer suggestion, these points have been treated.

After interpunction mark in many places it should be a blank space:

for example "Urtica dioica,often"

should be "Urtica dioica, often"

"become refractory or intolerant over time [13],and"

should be become refractory or intolerant over time, and  .... [13]

The interpunction marks and spaces are corrected in all manuscript.

New references are mentioned in the manuscript.

According to the reviewer suggestion, this point has been treated.

I think it requires some stylic, interpunction, and language editing.

All manuscript section has been checked for English language with the help of a scientific English user.

Round 2

Reviewer 1 Report

Comments and Suggestions for Authors

Thank you for revision

Comments on the Quality of English Language

OK